# Outcomes of excessive alcohol drinkers without baseline evidence of chronic liver disease after 15 years follow-up: Heavy burden of cancer and liver disease mortality

Sónia Bernardo[1], Ricardo Crespo[1], Sofia Saraiva[2], Rui Barata[3], Sara Gonçalves[4], Paulo Nogueira [5], Helena Cortez-Pinto [1,6], Mariana Verdelho Machado [1,6]*

1 Gastroenterology and Hepatology Department, Hospital de Santa Maria, CHULN, Lisbon, Portugal, 2 Nephrology Department, Hospital de Curry Cabral, CHULC, Lisbon, Portugal, 3 Gastroenterology Department, Portuguese Oncology Institute, Lisbon, Portugal, 4 Nephrology Department, Hospital de Santa Maria, CHULN, Lisbon, Portugal, 5 Biostatistics' Department, Faculdade de Medicina, Universidade de Lisboa, Lisbon, Portugal, 6 Clínica Universitária de Gastrenterologia, Faculdade de Medicina, Universidade de Lisboa, Lisbon, Portugal

* mverdelhomachado@gmail.com

**Data Availability Statement:** All relevant data are within the manuscript.

## Abstract

### Background

Most long-term heavy drinkers do not have clinically evident chronic liver disease (CLD). However, at any time-point, their risk of developing CLD remains unknown. We aimed to evaluate the long-term outcomes of a group of heavy drinkers, without evidence of CLD at baseline.

### Methods

A cohort of 123 long-term heavy drinkers without CLD were prospectively recruited in 2002 and retrospectively followed until 2018.

### Results

At baseline (2002), median alcohol consumption was 271±203g/day during 21.5±20 years, 65% being abstinent during the previous 1.75±5 months. Patients were followed for 14±3 years. During follow-up, 53% reported any alcohol intake. Alcohol consumption during follow-up associated weakly with either 1- or 6-months previous abstinence at baseline. Until 2018, progression to CLD occurred in 6%, associating with years of alcohol intake during follow-up (OR 1.15 [1.01–1.31]) and baseline alkaline-phosphatase (OR 1.05 [1.01–1.10]). During follow-up, being abstinent for at least 1 year positively associated with CLD-free survival. 27% died (55% of cancer–mostly oropharyngeal cancer, 27% of cardiovascular disease, and 9% of liver disease), with a mean age of 71 years [69–74] (10 years less than the expected in the Portuguese population). Achieving abstinence for at least 1 year positively associated with overall survival, while smoking, and hepatic steatosis at baseline associated negatively.

**Funding:** The authors received no specific funding for this work.

**Competing interests:** The authors have declared that no competing interests exist.

**Abbreviations:** ALT, alanine aminotransferase; ALP, alkaline phosphatase; AST, aspartate aminotransferase; BMI, body mass index; CLD, chronic liver disease; GGT, γ-glutamyl transferase; LC, liver cirrhosis.

## Conclusion

Long-term heavy drinkers seemed to have a decreased life expectancy compared with the overall Portuguese population. Cancer was the main cause of death. Our results suggest that progression to CLD depends mostly on continued alcohol intake. Alcohol abstinence, even if temporary, seems to decrease the risks of CLD and mortality.

## Introduction

Alcohol consumption is the third preventable cause of mortality, and the 7th cause of mortality, accounting for 5% of deaths worldwide [1]. Any amount of alcohol intake is associated with increased mortality [1]. Alcohol consumption is associated with more than 200 diseases [2], being the liver the single organ most commonly affected by excessive drinking [3]. Alcohol intake is responsible for up to 50% of cases of liver cirrhosis worldwide [4], with geographical differences: alcohol contributes to 15% of cirrhosis-related admissions in Africa, 48% in the US, and 72% in Europe [5]. However, only 15%-20% of heavy-drinkers will develop liver cirrhosis [2, 3]. The amount of alcohol intake seems critical in inducing liver disease. Indeed, there seems to be a dose-dependent effect between the amount of alcohol intake and the risk of developing chronic liver disease (CLD) [6], as well as, a threshold effect, with alcohol intake higher than 30g/day in men and 20g/day in women being considered potentially hepatotoxic [7]. The duration of alcohol consumption necessary to develop liver cirrhosis is unknown [8, 9] but there seems to be a linear increase with time. Development of liver cirrhosis has been described after just 4 years of alcohol consumption, however, the prevalence of cirrhosis increases exponentially after 8–12 years of consumption [10]. Several known co-factors increase the risk for alcohol-associated liver cirrhosis, such as hepatitis C virus infection, obesity, diabetes-mellitus, genetic factors (for example, polymorphisms in the PNPLA3 and TM6SF2 genes), smoking, and low coffee intake [3]. On the other hand, abstinence is known to decrease the risk for progression to cirrhosis, and to improve the prognosis of patients with cirrhosis [11, 12].

Little is known regarding the long-term outcomes of alcohol harmful consumers without known liver disease. Are those patients still at risk for developing CLD? We followed, for 15 years, a well-characterized cohort of harmful alcohol consumers, without evidence of CLD at baseline, in order to evaluate their long-term outcomes and to identify risk factors for mortality and further development of CLD.

## Material and methods

### Study design and patients

Patients with harmful alcohol consumption and evidence of alcohol-dependence, followed in an Alcohol Rehabilitation Clinic, were prospectively and consecutively enrolled in a clinical protocol performed in 2002, with data partially reported in 2005 [13], and retrospectively re-evaluated in 2018. At enrollment, all patients attending outpatient visits at *Centro Regional de Alcoologia do Sul* (CRAS) were asked to participate in the study and were evaluated at a Hepatology visit.

At baseline, inclusion criteria were: 1) alcohol consumption of at least 50 grams per day during at least 5 years, 2) age greater than 18 years, and 3) agreement to participate in the study. Abstinence at recruitment was accepted. Exclusion criteria were: 1) presence of liver

disease not related to alcohol (i.e. infection by hepatitis B/C virus, primary biliary cholangitis, autoimmune hepatitis, primary sclerosing cholangitis, Wilson's disease, hemochromatosis, or α1-antitrypsin deficiency), 2) treatment with potentially hepatotoxic drugs within 6 months of enrollment, and 3) active cancer. After thorough clinical evaluation by a dedicated Hepatologist, patients with clinical evidence of CLD or with altered liver blood tests were also excluded. CLD was defined by the presence of a past medical history of hepatic decompensation (jaundice, ascites, hepatic encephalopathy, and/or variceal bleeding), or histological confirmation of liver disease of greater severity than steatosis (liver biopsy was performed in 16 patients). Requisites for the absence of CLD included: the absence of a history of hepatic decompensation, absence of physical stigmata of chronic liver disease (i.e. cutaneous signs, hepatosplenomegaly, gynecomastia, and testicular atrophy), and at least two normal determinations of blood liver tests (aminotransferases, alkaline phosphatase, total bilirubin, prothrombin time and albumin) with an exception of an isolated rise in γ-glutamyl transpeptidase (GGT). Isolated increase of GGT was not an exclusion criteria, since it is a marker of alcohol consumption and not of liver injury, translating hepatic enzyme induction rather than liver cell injury [14]. Of the 262 patients screened, 130 did not have evidence of CLD, 103 had alcohol-associated liver disease, and the remaining 29 had either other liver diseases or mild increase in liver enzymes (that is, an increase lower than 3 times the reference value). Isolated increase of GGT was found in 39 patients (31.7%) without evidence of liver disease. Patients to whom there were clinical doubts regarding the presence of liver disease motivating the need for a liver biopsy were excluded. Those patients could not be classified as absence of liver disease (according to protocol) even if the biopsy was normal, because sampling error could not be excluded. Lastly, in the group of selected patients without evidence of CLD, we retrospectively applied the non-invasive score FIB-4 [15] to assess advanced fibrosis, and the ALBI-FIB4 score [17] to assess the risk of hepatic decompensation, at baseline.

Data were collected through an interviewer-administered semi-structured questionnaire including alcohol intake assessment, clinical examination, and blood tests.

All participants gave written informed consent, and the study was approved by the Ethics committee of the institutions involved: Comissão de Ética do Centro Académico de Lisboa—CAML in 2002 and again in 2018 (ID number 13/18).

## Baseline evaluation

**Demographics and alcohol consumption data.**   In 2002, the questionnaire included demographic data (age, gender, race) and details regarding alcohol intake. Patients were questioned about total duration of alcohol consumption, abstinence periods, beverage type (wine, beer, or spirits), and average daily amount of alcohol intake (expressed in grams of pure ethanol). The baseline questionnaire also included smoking habits, medication, and family history of alcohol abuse or liver disease.

**Clinical and biochemical assessment.**   A physical examination was performed including height (kg), weight (m), body mass index (BMI) (kg/m$^2$), and the presence of stigmata of alcohol consumption or CLD.

The following blood tests were performed: blood count and liver tests, namely alanine aminotransferase (ALT), aspartate aminotransferase (AST), alkaline phosphatase (ALP), GGT, bilirubin, albumin, and prothrombin time. To exclude other causes of CLD, viral serologies (hepatitis B and C), autoantibodies screen, iron metabolism, ceruloplasmin, and α1-antitrypsin serum levels were performed. Renal function, serum sodium and potassium, lipid profile, and serum fasting glucose were also collected. All were analyzed at the Hospital's Central Pathology Department, using standardized methods.

All patients had a previous abdominal ultrasound performed at the rehabilitation clinic, within the previous 6 months. The presence of hepatomegaly, steatosis, splenomegaly, and ascites were recorded.

### Follow-up

Patients were re-evaluated in 2018. The end of follow-up was the date of the last clinical interview or the date of death. All data were obtained from the computerized national health database (including hospital admissions or outpatient visits, primary care visits, blood tests, and imaging results), from consultation of the records in the Alcohol Rehabilitation Clinic at 2011, and through a survey performed by telephone interview to the patient or patient´s relatives. The parameters evaluated were: 1) co-morbidities such as diabetes-mellitus, dyslipidaemia, cardiovascular disease, or cancer; 2) assessment of alcohol intake to estimate lifetime alcohol exposure, abstinence periods, duration and quantification of alcohol intake and type of beverage, 3) medication; 4) diagnosis of liver cirrhosis or CLD, and the number of liver decompensations/hospitalizations; 5) mortality and cause of death.

Patients were considered to have developed CLD when they presented abnormal liver tests for more than 6 months, imaging findings compatible with CLD, or presented with hepatic decompensation (jaundice, ascites, hepatic encephalopathy, and/or variceal bleeding).

### Statistical analysis

The primary endpoint was the estimation of the risk for developing alcohol-related CLD and associated risk factors, in long-term harmful alcohol consumers without baseline CLD. The secondary endpoint was the evaluation of overall mortality, causes of death, and associated risk factors.

Continuous variables were expressed as median±IQR or mean±standard deviation (SD) depending on the absence or presence of a normal distribution (according to Kolmogorov-Smirnov test), respectively. Continuous variables with normal distribution were compared using the independent t-Student test. Categorical variables were described using frequencies and compared using the chi-square test. The independence of the associations of variables with the development of CLD and survival were assessed by multivariable logistic regression analysis. Kaplan-Meier survival curves and Cox-regression were used to assess the outcomes CLD free- and overall-survival. Analyses were performed using IBM Statistical Package for the Social Sciences (SPSS) software version 21. Two-tailed p values less than 0.05 were considered statistically significant.

## Results

### General characteristics and demographics of the study cohort

At baseline, 130 patients with harmful alcohol consumption without evidence of CLD were recruited from an alcohol rehabilitation clinic. Of those, 7 were lost for follow-up, and 123 were included in the analysis. One hundred and one were male (82%) and the mean age was 45±11years. The mean duration of follow-up was 14±3years (maximum 16 years). Although, at baseline, no patient had evidence of CLD, 11 (9%) had previous biochemical changes compatible with asymptomatic ambulatory alcoholic hepatitis. At the time of recruitment, 2002, transient elastography was still not available for clinical practice, and it was deemed unethical to perform a liver biopsy to confirm the absence of liver disease. To better characterize this population, we retrospectively applied FIB-4 [15], a well accepted noninvasive score for liver fibrosis assessment. FIB-4 >3.25 (cutoff suggesting significant fibrosis) was present in one

patient, and FIB-4 >1.45 (cutoff used to exclude fibrosis) was present in 9 patients. Interestingly, no patient with FIB-4 >1.45 progressed to CLD on follow-up. We also applied the ALBI-FIB-4 [16] score and all patients presented a low risk for liver disease decompensation (the mean score was -3.91±0.51, minimum -5.308 and maximum -2.359). Concerning family history, 26 patients (21%) reported a family history of CLD and 75 (61%) of alcohol use disorder. Eighty-one (66%) were smokers, 48 (46%) overweight/obese (28 obese, 23%) and 6 patients had type 2 diabetes mellitus (5%). The demographic and clinical characteristics are presented in Table 1.

## Evaluation at baseline

At inclusion, the median duration and amount of daily alcohol intake were 21.5±20years (5–70) and 271±203g/day (50–1470), respectively. 80 patients (65%) reported abstinence, for 1.75 ±5months. Considering the pattern of consumption, most patients (n = 96, 78.1%) consumed more than one type of beverage, 77% consumed wine (n = 95), 73% beer (n = 90), and 72% spirits (n = 89).

## Evaluation at the end of follow-up

At the end of follow-up, 65 (53%) patients reported some alcohol consumption during the period of follow up. Relapse of alcohol intake did not associate with gender, duration, or amount of alcohol intake previous to baseline, type of beverage, family history of alcoholism, or CLD. Relapse of alcohol consumption presented a very weak association with duration of abstinence at baseline (Table 2). A previous abstinence period of 6 months did not outperform 3 or 1-month previous abstinence.

During follow-up, the median period of abstinence was 10±15years (0–42), whereas the median duration and amount of alcohol intake were 1.5±12years (0–16) and 32±176g/day (0–470), respectively.

The most relevant comorbidities were: cardiovascular disease in 68 patients (55%), cancer in 30 (24%), and type 2 diabetes-mellitus in 19 (15%).

Development of type 2 diabetes-mellitus positively associated with age (52±11 *vs.* 44 ±11years, p = 0.004), and previous years of alcohol consumption (31±15 *vs.* 24±12years, p = 0.020), whereas inversely associated with baseline smoking (32% *vs.* 72%, p = 0.001). Of note, we did not have information regarding the smoking status on follow up. Type 2 diabetes-mellitus did not associate with alcohol amount, type of beverage consumption, being over-weight/obese, nor with the presence of liver steatosis on ultrasound. On multivariable analysis, the only identified independent risk factors for the development of diabetes-mellitus were baseline smoking (OR 0.181, 95%IC [0.521–0.063]) and age (OR 1.054, 95%IC [1.002–1.109]).

Development of cardiovascular disease positively associated with baseline BMI (26±4 *vs.* 24 ±3kg/m$^2$, p = 0.002), average daily alcohol intake during follow-up (158±276 *vs.* 85±11g, p = 0.050), and total alcohol intake during follow-up 643±137 *vs.* 261±411kg, p = 0.034). Cardiovascular disease negatively associated with liver tests at inclusion (GGT: 54±72 *vs.* 106 ±176IU/L, p = 0.042). On multivariable analysis, the only identified independent risk factor for the development of cardiovascular disease was BMI (OR 1.181, 95%IC [1.054–1.323]).

During follow-up, 30 patients developed cancer: 11 oropharyngeal, 6 gastrointestinal, 3 urothelial, 3 lung, 2 breast, 2 skin, 1 prostate, 1 leiomyosarcoma, and 1 leukemia. Cancer development positively associated with the following variables at inclusion: age (49±10 *vs.* 44±11years, p = 0.015), smoking (83% *vs.* 60%, p = 0.026), presence of alcoholism stigmata (90% *vs.* 72%, p = 0.049) namely facial angiectasias (48% *vs.* 25%, p = 0.036), and presence of liver steatosis on ultrasound (87% *vs.* 51%, p = 0.010). On multivariable analysis, the identified independent risk

**Table 1. Patient characteristics at inclusion.**

| | All Patients | No liver events on follow-up | Liver events on follow-up | p |
|---|---|---|---|---|
| Patients (N) | 123 | 116 | 7 | |
| Age (years)* | 45.5±11 | 45±11 | 52±9 | 0.124 |
| Male sex | 101 (82%) | 95 (83%) | 5 (71%) | 0.607 |
| Body mass index (kg/m²)* | 25±4 | 25±4 | 25±4 | 0.982 |
| Obese/overweight | 56 (46%) | 53 (46%) | 3 (43%) | 0.849 |
| Presence of alcohol stigmata | 93 (76%) | 100 (86%) | 6 (86%) | 1.000 |
| Smoking habits | 81 (66%) | 75 (65%) | 6 (86%) | 0.420 |
| Family history of alcoholism | 75 (61%) | 74 (64%) | 7 (100%) | 0.096 |
| Family history of liver disease | 28 (23%) | 25 (22%) | 2 (33%) | 0.618 |
| Alcohol intake per day (g) | 271±203 | 286±207 | 237±192 | 0.449 |
| Years of alcohol intake | 21.5±20 | 20±18 | 38±36 | 0.423 |
| Total alcohol intake (tons) | 2.5±2.4 | 2.4±2.4 | 2.6±2.7 | 0.982 |
| Type of beverage consumed: | | | | |
| Wine | 95 (77%) | 88 (76%) | 7 (100%) | 0.349 |
| Beer | 90 (73%) | 86 (74%) | 4 (57%) | 0.384 |
| Spirituous drinks | 89 (72%) | 85 (73%) | 4 (57%) | 0.394 |
| Multiple types | 96 (78%) | 90 (78%) | 5 (71%) | 0.666 |
| Months of alcohol abstinence | 1.75±5 | 1.5±5 | 2±3 | 0.703 |
| Hepatic steatosis on ultrasound | 74 (60%) | 70 (60%) | (5) 71% | 1.000 |
| Previous pattern of ambulatory alcoholic hepatitis | 11 (9%) | 10 (9%) | 1 (14%) | 0.490 |
| Laboratorial data: | | | | |
| Mean corpuscular volume (fL)* | 94±6 | 94±6 | 97±6 | 0.274 |
| Platelet count (cells/uL) | 239500±76750 | 237000±78000 | 271000±82000 | 0.423 |
| Alanine aminotransferase (IU/L) | 15±8 | 15±7 | 18±18 | 0.842 |
| Aspartate aminotransferase (IU/L) | 17±13 | 17±13 | 17±12 | 0.284 |
| γ-glutamyl transpeptidase (IU/L) | 32±63 | 32±58 | 41±81 | 0.912 |
| Alkaline phosphatase (IU/L) | 73±27 | 72±24 | 113±60 | **0.078** |
| Total bilirubin (mg/dL) | 0.4±0.2 | 0.4±0.2 | 0.3±0.3 | 0.704 |
| Albumin (g/L)* | 43±4 | 43±4 | 41±2 | 0.123 |
| INR* | 0.9±0.1 | 0.9±0.1 | 0.9±0.1 | 0.174 |
| Total cholesterol (mg/dL)* | 211±44 | 212±44 | 189±45 | 0.182 |
| Triglycerides (mg/dL) | 125±105 | 126±104 | 125±123 | 0.982 |
| Urea (mg/dL) | 29±14 | 29±12 | 22±11 | **0.031** |
| Creatinine (mg/dL) | 0.7±0.2 | 0.7±0.2 | 0.6±0.1 | 0.103 |
| Glucose (mg/dL) | 80±20 | 81±22 | 72±17 | 0.910 |
| Non-invasive scores of fibrosis | | | | |
| FIB-4 >1.45 | 9 (7%) | 9 (8%) | 0 (0%) | 0.444 |

Data is expressed as mean±standard deviation (*), when variables presented a normal distribution or median±IQR when variables did not present a normal distribution (according to Kolmogorof-Smirnov test).

factors for cancer development were facial angiectasias (OR 6.99, 95%IC [1.71–28.57]), smoking (OR 6.6.7, 95%IC [1.34–33.33]), and liver steatosis (OR 6.41, 95%IC [1.14–35.71]).

## Risk factors for the development of CLD

**CLD development.** During follow-up, 7 patients (6%) progressed to CLD, of whom 5 presented with liver decompensation. Clinical events were ascites in 4 (57%), hepatic

**Table 2. Duration of alcohol abstinence as a predictor of alcohol relapse.**

|  | Se | Sp | PPV | NPV | OR with 95%CI, p |
|---|---|---|---|---|---|
| > 1 month | 46% | 26% | 70% | 58% | 0.306 [0.142–0.658], p = 0.004 |
| > 3 months | 31% | 44% | 64% | 61% | 0.347 [0.165–0.729], p = 0.008 |
| > 6 months | 14% | 63% | 61% | 41% | 0.276 [0.114–0.668], p = 0.007 |

Se, sensitivity; Sp, specificity; PPV, positive predictive value; NPV, negative predictive value; OR, odds ratio.

encephalopathy in 1 (14%), and esophageal varices-related upper gastrointestinal bleeding in 1 (14%). Other complications of CLD such as spontaneous bacterial peritonitis, refractory ascites, hepatic hydrothorax, hepatorenal syndrome, or hepatocellular carcinoma, were not reported.

Progression to CLD associated with alcohol consumption during follow-up, but not with alcohol consumption previous to inclusion or total burden of alcohol intake. Indeed, progression to CLD was positively associated with years of consumption during follow up (14±12 *vs.* 5 ±7years, p = 0.002), and inversely with abstinence >1 year during follow up (43% *vs.* 79%, p = 0.047). The total lifelong duration of heavy alcohol intake was 46±21 with a minimum of 16 years and a maximum of 78 years. In this population, heavy alcohol intake for longer than 50 years had a good accuracy for the development of CLD, as 9 in 10 patients with this duration of consumption developed CLD (57% sensitivity and 89% specificity) (Fig 1).

Very early markers of liver disease might be higher ALP (113±60 *vs.* 72±24IU/L, p = 0.078) and lower urea (22±11 *vs.* 29±12mg/dL, p = 0.031), since both associated with progression to CLD. In multivariate analysis, the independent risk factors for progression to CLD were years of alcohol consumption during follow-up (OR 1.15, 95%IC [1.01–1.31]) and ALP (OR 1.05, 95%IC [1.01–1.10]).

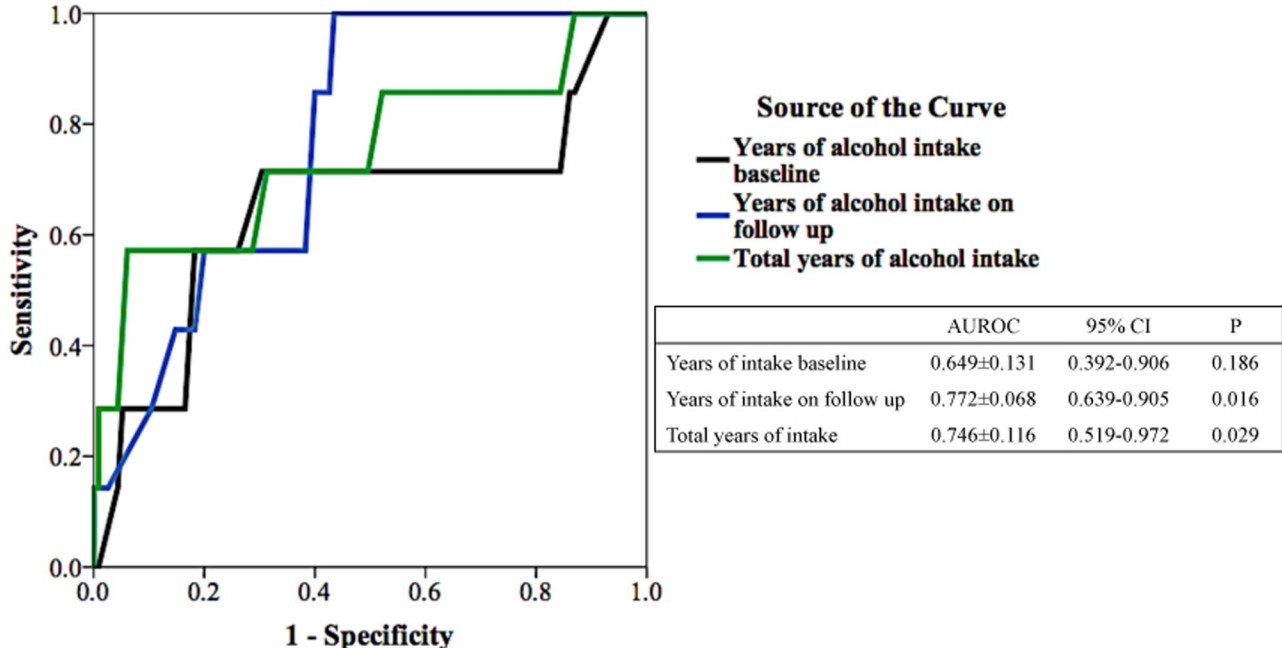

**Fig 1. Duration of alcohol intake and development of chronic liver disease.**

**CLD-free survival.** In order to allow an analysis of time to progression, Kaplan Meier curves were computed. CLD-free survival was not affected by the amount or duration of alcohol consumption before baseline. However, achieving at least one year of abstinence during follow up associated with higher CLD-free survival (15.6±0.2 *vs.* 14.7±0.6 years, p = 0.019) (Fig 2). The hazard ratio for CLD-free survival, after Cox regression analysis, was 3.01 [0.65–13.97] (p = 0.159) for achieving at least one year of abstinence during follow up, 1.04 [1.01–1.08] (p = 0.004) for ALP at baseline, and 0.87 [0.77–0.99] (p = 0.039) for urea at baseline.

## Risk factors for mortality

During follow-up, 33 patients (27%) died. The causes of death were cancer (n = 18, 55%), cardiovascular diseases (n = 9, 27%), liver disease (n = 3, 9%), car accident (n = 2, 6%), and pneumonia (n = 1, 3%) (Fig 3).

The mean age at death was 71±1years (41–77). Factors associated with longer survival were abstinence for at least one year during follow-up (73±1 *vs.* 65±2years, p = 0.013) and beer consumption at baseline (74±1 *vs.* 66±2years, p = 0.001). On the contrary, factors associated with decreased survival were smoking (69±2 *vs.* 74±1, p = 0.017) and hepatic steatosis on baseline ultrasound (68±2 *vs.* 75±2, p = 0.015) (Fig 4).

## Discussion

We present a 15-year longitudinal study on long-term heavy-drinkers without CLD. These patients represent the largest subgroup of heavy-drinkers on clinical practice (particularly in Primary Care Clinics), however, little is known regarding their real risk of progression to CLD and their co-morbidities [3, 17]. Many patients who died from alcohol-related CLD had previous non-liver related recurrent admissions to the hospital, which might have been missed opportunities for early intervention [18]. As such, it is pressing to better understand the prognosis of these patients and to identify early predictors of progression to CLD.

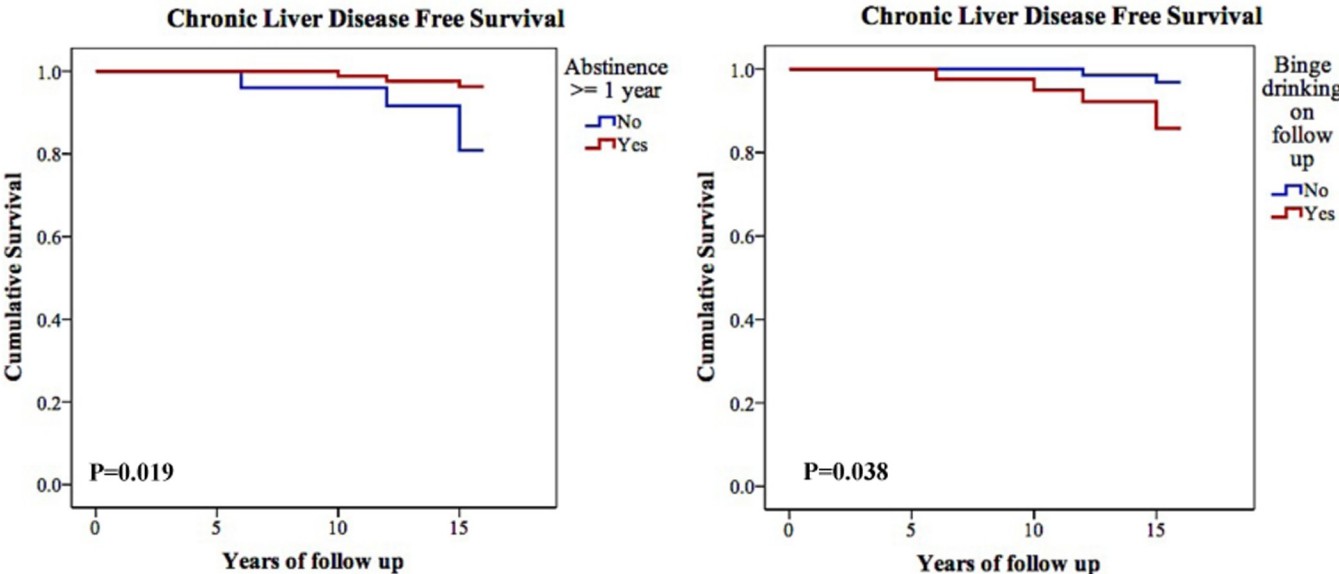

**Fig 2. Alcohol intake pattern on follow-up modulates chronic liver disease-free survival.** Kaplan-Meier curves showing chronic liver disease-free survival according to the abstinence of at least one year and binge-drinking on follow-up. The log-rank test was used to compare the two curves.

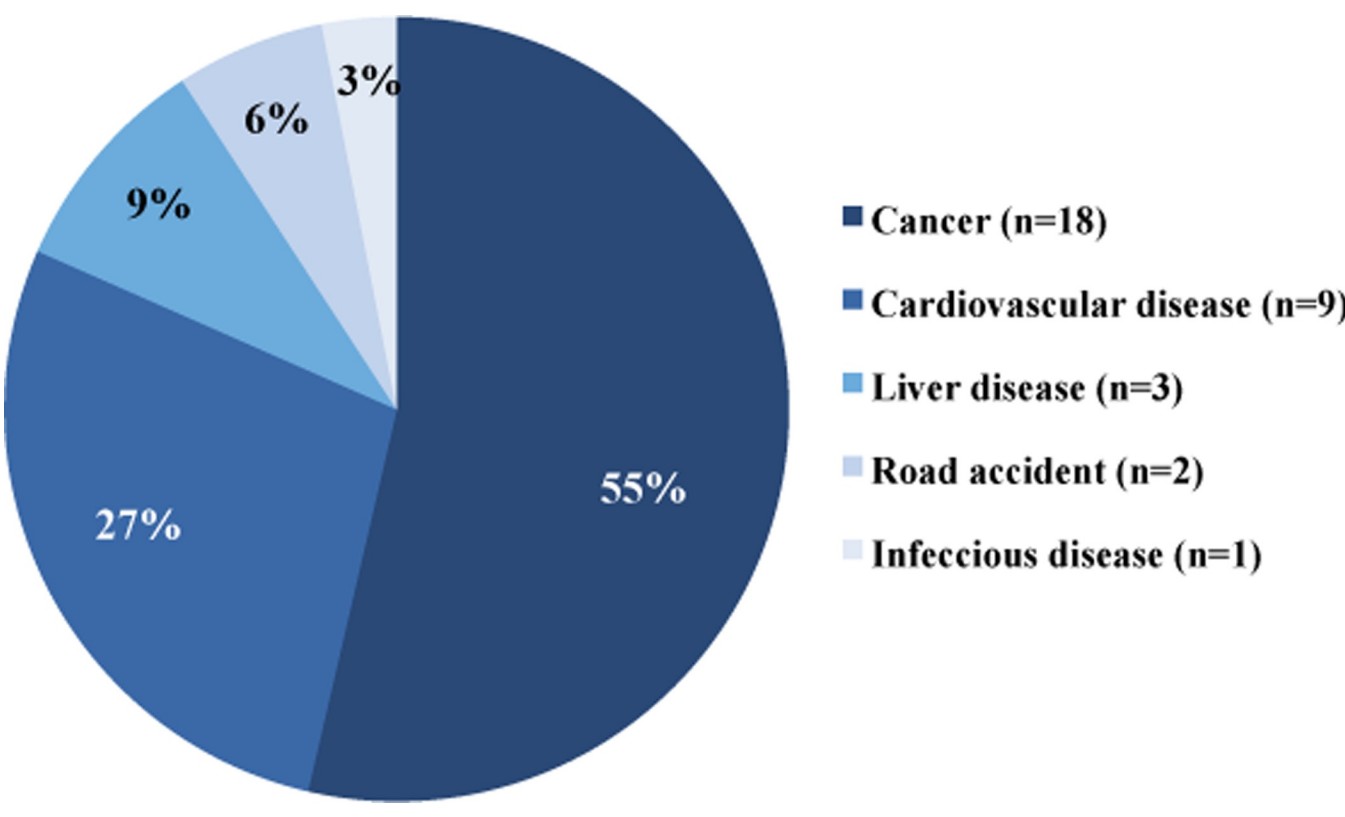

**Fig 3. Causes of death.**

The first striking finding, from this cohort, is that patients who survived up to 25 years of heavy-drinking without developing evidence of CLD, and hence who were more likely resistant to the dismal effects of alcohol on the liver, are still at increased risk for CLD. 6% of those patients did progress to CLD during a follow-up of 15 years. Interestingly, lifelong heavy alcohol intake for 50 years seems to almost invariably result in CLD, since 9 in 10 patients with this long-term intake did progress to CLD. Furthermore, in this cohort, liver-related mortality was the third cause of death, representing 9% of all deaths, which is 10-times higher than the burden of liver death in Portugal (0.9%) [19].

When we evaluate a heavy drinker without CLD, the previous amount, duration, and type of alcohol intake do not seem to predict progression to CLD. The risk of developing CLD seems to be determined by the forthcoming consumption of alcohol. Indeed, for each year of heavy alcohol intake during follow-up, the estimated risk of CLD increased by 15%, compared to patients who remained abstinent. A large Danish study also corroborated that recent drinking (i.e. in the previous decade) seems to better predict progression to CLD, as compared to earlier drinking [20]. Similarly, achieving abstinence, even if temporarily, for as short as one year, strongly reduced the risk for CLD and overall mortality. As such, this study highlights that interventions to promote abstinence is highly recommended, and that even small achievements can impact the long-term prognosis of harmful alcohol consumers.

Regarding the controversial topic of the type of alcoholic beverages consumption as a risk factor for CLD [20–25], no association was found in these patients. Furthermore, in the heavy drinking range, the amount of alcohol consumed did not seem to impact the risk for CLD. We did not have information regarding binge-drinking.

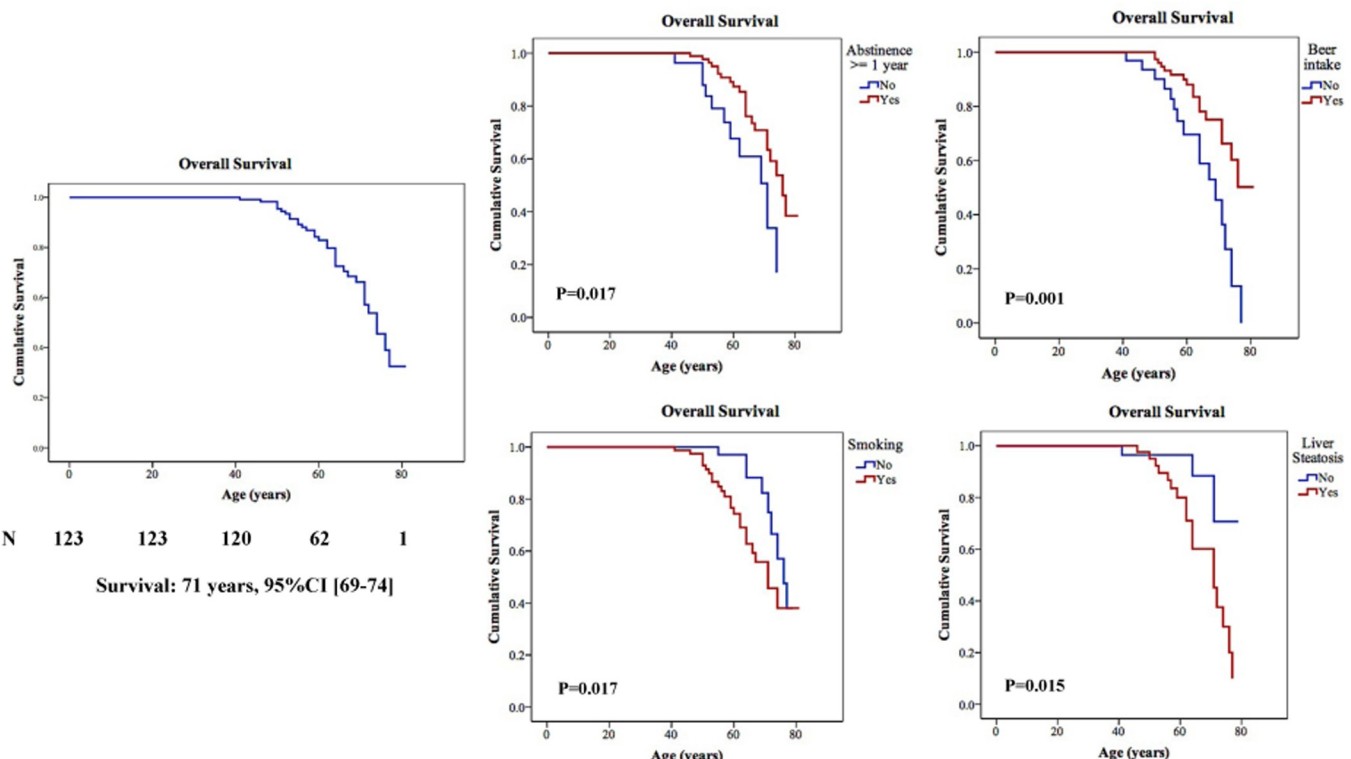

**Fig 4. Factors associated with overall survival.** Kaplan-Meier curves showing the overall survival according to alcohol intake pattern, smoking habits, and presence of liver steatosis on ultrasound. The log-rank test was used to compare the two curves.

This cohort suggests that blood urea and ALP might be very early markers of liver disease. Since urea is produced in the liver from the metabolism of amino-acids, it is possible that even in the very early stages of liver disease, levels and activity of urea cycle enzymes are decreased [26] leading to decreased synthesis of urea [27]. Furthermore, mild ALP elevation might represent subclinical intrahepatic cholestasis, a known histological feature associated with a worse prognosis in alcohol-related liver disease [11, 28].

In recent literature, a strong association has been suggested between metabolic factors such as obesity and type 2 diabetes-mellitus and increased risk of CLD [29–33], which did not occur in this cohort.

Also of note was the heavy burden of cancer in these patients. Cancer was the number one cause of mortality, responsible for more than half of the cases of death, which is about two times higher the expected in the Portuguese population [19]. The main cancer was oropharyngeal cancer, which, as expected [21, 25, 30], showed a strong association with smoking. This is of concern since drinkers are more likely to smoke than nondrinkers [34, 35]. Further studies should address if this population should be submitted to a personalized screening program.

Finally, the mean age of death was 71 years, which is 10 years younger than the reported life expectancy in Portugal [36]. This result goes in agreement with a cohort study with 19002 alcohol-dependent individuals from Denmark, which showed a potential loss of 10 years of life when compared with 186767 controls from the Danish Civil Registration System [37]. Interestingly, our cohort had a lower rate of progression (6% *vs.* 25%) than the Danish cohort, probably related to the fact that these were patients without evidence of CLD baseline, suggesting a significant degree of resistance to liver disease development.

Similarly, a recent Sweden population-based cohort study on 3453 patients who performed liver biopsy for suspicion of alcohol-related liver disease, showed that patients with normal liver histology or simple steatosis, presented a 2–3 fold increased mortality compared to the matched general population, particularly if continuing heavy drinking [38].

The strongest factors associated with mortality were smoking and hepatic steatosis. Of note, baseline hepatic steatosis was associated with overall mortality, even though it did not associate with the liver outcome, cardiovascular diseases, or diabetes-mellitus. However, hepatic steatosis did associate with cancer, which might suggest that alcohol-related liver steatosis might induce a systemic procarcinogenic environment [39]. Once again, alcohol abstinence, for as short as one year, showed a positive impact on survival.

The study strengths include its prospective design, the long duration of follow-up, the extensive data collection, and the evaluation of lifetime alcohol exposure. Furthermore, all patients were initially assessed by a Hepatologist, and hence we are confident regarding the exclusion of liver disease at baseline, and in our ability to control for several potential confounders. To strengthen the confidence of exclusion of CLD, we retrospectively applied noninvasive scores of fibrosis, and only one patient presented FIB-4 suggestive of advanced liver fibrosis. Furthermore, we applied the ALBI-FIB4 score and no patient presented a high-risk score for liver decompensation. Still, the current study has some limitations. First, the risks of bias due to the nature of alcohol history questionnaires with recall issues. Indeed, we tried to be thorough in the evaluation of alcohol consumption, applying a semi-structured questionnaire to patients and their relatives, while confronting them with the information collected from medical records from the national database, as well as information collected from the medical records from the alcohol rehabilitation clinical at the middle of the follow up. Second, liver biopsies confirming the absence of CLD at baseline and on follow-up were not performed, as it would be deemed unethical to submit healthy subjects to an invasive procedure. Furthermore, transient hepatic elastography was also not performed because it was not accessible at the time of inclusion of the patients. However, liver biopsy is not routinely performed in these patients in the real-life clinical practice, and many centers (mainly in the primary care set) still do not have easy access to transient hepatic elastography. Thus this study may help clinicians to guide evidence-based clinical decisions in their daily practice, in patients without evidence of CLD by conventional diagnostic approaches. Third, the development of CLD during follow-up was a rare event, thus making the comparisons weaker. We also included a small number of women. Lastly, the analysis of the effect of variables and confounders on outcomes has the risk of bias by the lack of control to time to progression, which was taken into account in the evaluation of the survival curves.

In conclusion, this study showed that even long-term heavy-drinkers with no evidence of CLD, and periods of abstinence, can still progress to end-stage liver disease, which depends more on forthcoming alcohol intake rather than previous alcohol intake. Furthermore, achieving alcohol abstinence, even for a short period of time, has a positive impact on the risk of progression for CLD and mortality.

In conclusion, for clinical practice, our results suggest that special attention should be made regarding the risk for cancer, with personalized screening programs and intensive anti-smoking interventions, and also those periods of alcohol abstinence are paramount, even with interspersed periods of relapse.

## Author Contributions

**Conceptualization:** Helena Cortez-Pinto, Mariana Verdelho Machado.

**Data curation:** Sónia Bernardo, Ricardo Crespo, Sofia Saraiva, Rui Barata, Sara Gonçalves, Mariana Verdelho Machado.

**Formal analysis:** Paulo Nogueira, Mariana Verdelho Machado.

**Supervision:** Helena Cortez-Pinto, Mariana Verdelho Machado.

**Writing – original draft:** Mariana Verdelho Machado.

**Writing – review & editing:** Sara Gonçalves, Paulo Nogueira, Helena Cortez-Pinto, Mariana Verdelho Machado.

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
