## [Decision Letter · Decision Letter 0]

23 Mar 2021

PONE-D-21-05110

Outcomes of excessive alcohol drinkers after 15 years follow-up: heavy burden of cancer and liver disease mortality

PLOS ONE

Dear Dr. Machado,

Thank you for submitting your manuscript to PLOS ONE. After careful consideration, we feel that it has merit but does not fully meet PLOS ONE’s publication criteria as it currently stands. Therefore, we invite you to submit a revised version of the manuscript that addresses the points raised during the review process.

Special attention should be given to the issue of retrospective evaluation of alcohol drinking history during long-term follow-up: As pointed out by Reviewers 1 and 3, more details on drinking history should be provided or - if those are lacking - this limitation should be stated more clearly in the Discussion. 

I concur with Reviewer 2 that ROC analysis does not seem appropriate for dichotomous variables (Figure 1).

We look forward to receiving your revised manuscript.

Kind regards,

Rudolf E. Stauber, MD

Academic Editor

PLOS ONE

Journal Requirements:

2. Thank you for including your ethics statement:  "This study was approved by the CHULN and CAM ethic committee (ID number 13/18). All patients gave their informed, written consent to participate in the study.".   

3. Please include additional information regarding the survey or questionnaire used in the study and ensure that you have provided sufficient details that others could replicate the analyses. For instance, if you developed the survey or questionnaire as part of this study and it is not under a copyright more restrictive than CC-BY, please include a copy, in both the original language and English, as Supporting Information. If the questionnaire is published, please provide a citation to the (1) questionnaire and/or (2) original publication associated with the questionnaire.

4. Please provide the name of the alcohol rehabilitation clinic.

Reviewers' comments:

Reviewer's Responses to Questions

**Comments to the Author**

1. Is the manuscript technically sound, and do the data support the conclusions?

Reviewer #1: Partly

Reviewer #2: Yes

Reviewer #3: Partly

2. Has the statistical analysis been performed appropriately and rigorously? 

Reviewer #1: Yes

Reviewer #2: No

Reviewer #3: No

3. Have the authors made all data underlying the findings in their manuscript fully available?

Reviewer #1: No

Reviewer #2: Yes

Reviewer #3: No

4. Is the manuscript presented in an intelligible fashion and written in standard English?

Reviewer #1: Yes

Reviewer #2: Yes

Reviewer #3: No

5. Review Comments to the Author

Reviewer #1: I consider this an important study since alcohol-related health issues are significant and we lack enough data on long-term follow up in heavy drinkers. In my opinion, since it is quite challenging to enroll drinkers in such studies, I am fine with the partly retrospective aspect of the study and partly missing data. It is also natural that transient elastography was not available. However, the real and important data should be shown, and the retrospective study design should be visible already in the abstract. An important limitation of this retrospective study design is the diagnosis of “chronic liver disease” (CLD), since it cannot actually be ruled out based on the used conventional diagnostic approaches. Both lab tests and routine ultrasound can overlook manifest liver cirrhosis in up to 50%. Pending a critical revision, the paper could be considered again for publication since it provides important clinical read outs in a long-term followed up cohort of heavy drinkers.

Major comments/questions:

1. It is correct that we do not know enough about heavy drinkers outcome with no liver disease. But the absence of CLD cannot be done efficiently based on the diagnostic criteria as mentioned above. In addition, APRI score, although quite usefuly in the setting of viral hepatitis, performs very poorly in ALD. IN contrast, FIB4 can be accepted, and I recommend to solely stratify the cohort based on FIB4 since no other data such as ELF are available. Fib4 could also be combined with the available albumin since the recently introduced Albi Fib4 score seems to perform even better.

2. The strength of the study is the rather careful medical history and the descriptions of the various causes of death. I miss a clear description of these findings either in supplemental table or even in the major text body.

3. The risk to develop ALD is described to be as high as 15% per year. This seems to by far to highy. E.g. in other cohorts of heavy drinkers, 20% developed cirrhosis after 15 years of heavy drinking.

4. Baseline data: Again it is not very clear but important for this study how patients were enrolled: It reads that “Patients with harmful alcohol consumption and evidence of alcohol-dependence, followed in an Alcohol Rehabilitation Clinic, were prospectively and consecutively enrolled in a clinical protocol performed in 2002 ref 13 and retrospectively re-evaluated in 2018. What means retrospectively in this context? How was liver disease excluded? I am also confused when looking at Ref 13: Herewith some data:

5. Ref 13 (Martins A et al Eur J Gastroenterol Hepatol 2005; 17 (10):1099-1104.). In this study, completely different patient numbers are provided

6. It is also interesting that obviously an an isolated rise in GGT was not considered an exclusion criterium. Why? How many patients were enrolled based on these criterium.

7. It is further pointed out that follow up data were retrospectively obtained form a national database. IT would be interesting to learn how data on alcohol consumption are regularly entered in this data base. Who is doing this? How often did patients undergo alcohol detoxification?

8. Since alcohol drinking history is essential for this study, more details should be provided, given the fact that it is generally challenging for patients to provide a detailed alcohol history with stops and goes. How was duration of alcohol consumption assessed. How do patients remember an alcohol drinking history of the last 10 years or so? In line with this, a 53% of relapse appears to be rather low for heavy drinkers

9. On page 12 it is stated that 9% had previous AH while in the inclusion criteria, previous decompensation was ruled out?

10. What about the numbers of diabetes and obesity?

11. Smoking is shown as risk factor for DM II with an OR 0.181 Does smoking protect from diabetes? There are studies continueosly showing a tight association between smoking and alcohol consumption? What are the numbers for this cohort?

Reviewer #2: This is an interesting prospective study that studies the natural history of heavy drinkers without evidence of liver disease and evaluates their clinical outcomes, risk of death and of developing liver disease during 15 years of follow up. Risk factors were also identified.

From this standpoint, the study is important because little is know about the risk of developing CLD in heavy drinkers. However, there are some issues that need further clarification:

- the data in table 1 are presented as mean±SD, which - based on the figures, seem not quite appropriate (at least for some of the variables. Could you please look back to the distribution of the variables and use median±IQR/range when needed. Also, in some cases, non parametric tests should have been used.

- regarding the data presented in figure 1, I think that the AUROC analysis is not the right choice for what was intended by the authors. AUROC is usually used with continuous variables, not with dichotomial ones. I believe a regression analysis (or a basic chi-square test) would be better choices.

- the manuscript also need a further English polish ti increase its readability.

Reviewer #3: Mariana Machado and colleagues present a single-center study from Lisbon on the natural history of 130 patients with a history of excessive drinking who present at an alcohol rehabilitation unit, but without evidence of chronic liver disease or cancer. The authors are commended for managing to recall almost all (7 losses to follow up) participants after 14±3 years for repeated investigations and a detailed description of alcohol use during follow up.

While there is a need for longitudinal studies on the natural history of patients with harmful drinking, taking fluctuations in alcohol use over the years into account, the study in its current form has substantial methodological and reporting issues which limits interpretation of results. Please find below my questions, comments and suggestions for improvement.

MAJOR:

• Please use the STROBE guidelines for reporting: https://www.equator-network.org/reporting-guidelines/strobe/

• Title: the short titles accurately states that development of CLD during follow up is the outcome of interest, while the long title does not.

• Lay summary, first point. It is not surprising that ongoing rather than past alcohol intake dictates the risk of liver cirrhosis development when you specifically exclude anyone with evidence of liver disease at baseline. Therefore I suggest to delete this sentence.

• The authors stress the point that alcohol consumers on average has a 10 yr reduced life expectancy. Their way of comparing cohort data with a country average is not really methodologically correct – the correct comparator would be 5:1 or 10:1 gender, age, time and geography matched random sample from the population. Therefore, I would propose not to emphasize the 10 yr reduction in life expectancy as much as the authors do – e.g. it appears in both abstract and bullet points.

• Introduction: The statement “Liver disease is the main alcohol-related chronic illness” gives the wrong impression, since liver disease – as the authors show – does not account for the majority of alcohol-related harm. I would propose instead to stress that the the liver is the single organ most commonly affected by excess drinking.

• Methods: Reference 13 is referred to as the original study. However, it describes a cohort of only 76 patients without evidence of liver disease at baseline, but this manuscript includes n=130 ?

• Methods: Please describe in more detail the in- and exclusion criteria. Which liver blood tests were considered? If fibrosis stage F1 on liver biopsy, the patient were considered to have ALD and excluded? From the text is seem like a very heterogeneous cohort, since many with compensated cirrhosis at baseline may have normal liver blood tests (see e.g. Table 1 in Mueller, World J Gastroenterol 2014 October 28; 20(40): 14626-14641). Those patients would be included unless they had a liver biopsy? In contrast, many with no or minimal fibrosis may have elevated transaminases and alkaline phosphatase – they would be excluded?

• Methods: Similarly, the definition of the primary outcome, development of CLD, is poorly described. The current description can encompass patients with anything from steatohepatitis without fibrosis, all the way to decompensated cirrhosis.

• Methods: The authors state that inclusions were in 2002 and reassessment in 2018, which would result in 16 years between baseline and reevaluation. Yet, in methods they state a follow up of 14 ±3 yrs?

• Clearly, the alcohol habits of those who die in between baseline and follow up can not be evaluated, or did the authors have information from other sources on alcohol behaviour during follow up?

• Methods: When the primary endpoint is risk of CLD, the authors should account for how they handle non-CLD death as a competing risk?

• Methods: Baseline CLD and cancer are excluded, but cardiovascular or metabolic disease are not, so the comparison between deaths from CLD, cancer and CVD is unbalanced.

• Results: seven patients (6%) progressed to CLD, which is lower than in studies of similar populations (for example Holst, Addiction 2017, vol 112, where approximately 25% die from liver related death). Could the authors discuss this? Would more accurate liver tests – like FibroScan, which was widely available in 2018 – have resulted in higher proportion being detected with CLD?

• Results: It makes less sense to discuss biomarkers of liver disease, when abnormal liver blood tests were an exclusion criterium.

• Discussion: It is problematic that there is such a long time period between baseline and follow up with regards to obtaining alcohol history. I fear that recall bias may play a strong role, since those who have developed CLD will be more prone to remember alcohol drinking periods than those who have not. Can the authors discuss this limitation?

• I recommend a language revision by a native english speaker. Examples from the abstract: “Most long-term heavy drinkers do not present chronic liver disease” (present with CLD?); “being abstinent for at least 1 year positively modeled CLD-free survival” (correlated or associated with CLD-free survival?); “showed an impressive decreased life expectancy” (impressive has positive connotations, which is in stark contrast to decreased life expenctancy).

6. PLOS authors have the option to publish the peer review history of their article (what does this mean?). If published, this will include your full peer review and any attached files.

Reviewer #1: No

Reviewer #2: No

Reviewer #3: No

---

## [Author Response · Author response to Decision Letter 0]

30 Mar 2021

We thank the editor and the reviewers for the constructive critiques and for the opportunity to improve our manuscript. We hope the reviewers now find it worthy for publication. It follows the point-by-point response to the editors and reviewers comments.

EDITORS’ COMMENTS

We performed the necessary changes.

2. Thank you for including your ethics statement: "This study was approved by the CHULN and CAM ethic committee (ID number 13/18). All patients gave their informed, written consent to participate in the study.". 

We added to the methods the following: “Comissão de Ética do Centro Académico de Lisboa _ CAML in 2002 and again in 2018.”

3. Please include additional information regarding the survey or questionnaire used in the study and ensure that you have provided sufficient details that others could replicate the analyses. For instance, if you developed the survey or questionnaire as part of this study and it is not under a copyright more restrictive than CC-BY, please include a copy, in both the original language and English, as Supporting Information. If the questionnaire is published, please provide a citation to the (1) questionnaire and/or (2) original publication associated with the questionnaire.

We did not use a specific questionnaire, rather we performed a detailed clinical history in an outpatient Hepatology visit.

4. Please provide the name of the alcohol rehabilitation clinic.

We added to the methods the following sentence: “At enrollment all patients attending outpatient visits at Centro Regional de Alcoologia do Sul (CRAS) were asked to participate in the study and evaluated at a Hepatology visit with laboratorial evaluation.”

We removed that information from other sections besides the Methods.

REVIEWER 1 

I consider this an important study since alcohol-related health issues are significant and we lack enough data on long-term follow up in heavy drinkers. In my opinion, since it is quite challenging to enroll drinkers in such studies, I am fine with the partly retrospective aspect of the study and partly missing data. It is also natural that transient elastography was not available. However, the real and important data should be shown, and the retrospective study design should be visible already in the abstract. An important limitation of this retrospective study design is the diagnosis of “chronic liver disease” (CLD), since it cannot actually be ruled out based on the used conventional diagnostic approaches. Both lab tests and routine ultrasound can overlook manifest liver cirrhosis in up to 50%. Pending a critical revision, the paper could be considered again for publication since it provides important clinical read outs in a long-term followed up cohort of heavy drinkers.

We thank the reviewer for these encouraging words. We agree that the ruling out of “chronic liver disease” has flaws, but we consider that we have done as best as possible at that time to achieve it.

Major comments/questions:

1. It is correct that we do not know enough about heavy drinkers outcome with no liver disease. But the absence of CLD cannot be done efficiently based on the diagnostic criteria as mentioned above. In addition, APRI score, although quite usefully in the setting of viral hepatitis, performs very poorly in ALD. IN contrast, FIB4 can be accepted, and I recommend to solely stratify the cohort based on FIB4 since no other data such as ELF are available. Fib4 could also be combined with the available albumin since the recently introduced Albi Fib4 score seems to perform even better.

We understand the reviewer considerations regarding the exclusion of chronic liver disease. We do agree with the limitations of conventional diagnostic approaches to rule it out. We go even further, since even the invasive gold standard evaluation with liver biopsy can misclassify liver fibrosis in up to on third of the patients (due to sample error and inter pathologist variability). However, the conventional diagnostic approaches mirrors real-life medical practice, and it may be the most useful information to help guiding evidence-based clinical decisions that clinicians have to make in their daily practice. Having said that, we were very thorough in our evaluation. An expert in Hepatology with special interest in alcohol-associated liver disease evaluated all patients. We needed two normal blood tests and ultrasound (except for isolated steatosis). Finally we retrospectively applied fibrosis scores, which confirmed low probability of advanced fibrosis in all patients. As such, we believe that we are the most confidants we could be regarding exclusion of liver disease, with the available tools at the time of enrollment (2002).

We followed the reviewer suggestion and removed the data on APRI and applied the ALBI-FIB4 score, and all patients presented a low risk score for liver decompensation. We added the following sentence in the methods: “We also applied the ALBI-FIB-4 score and all patients presented a low risk of decompensation of liver disease (the mean score was -3.91±0.51, minimum -5.309 and maximum -2.359).

2. The strength of the study is the rather careful medical history and the descriptions of the various causes of death. I miss a clear description of these findings either in supplemental table or even in the major text body.

We thank the reviewer for this suggestion. Accordingly, in the results, we changed the sentence: “The 3 main causes of death were cancer (n=18.55%), cardiovascular diseases (n=9.27%), and liver disease (n=3. 9%) (Figure 4).” to “The causes of death were cancer (n=18, 55%), cardiovascular diseases (n=9, 27%), liver disease (n=3, 9%), car accident (n=2, 6%), and pneumonia (n=1, 3%) (Figure 4).”

3. The risk to develop ALD is described to be as high as 15% per year. This seems to by far to highy. E.g. in other cohorts of heavy drinkers, 20% developed cirrhosis after 15 years of heavy drinking.

We thank the reviewer for this comment, as we were not clear conveying the message. Only 6% progressed to clinically evident chronic liver disease after 15 years of follow up. Compared to patients who remained abstinent, the risk of developing chronic liver disease increased 15% per year of alcohol consumption on follow up.

In the lay summary, we added to the sentence “For each year of heavy alcohol intake during follow-up, the risk of developing chronic liver disease increased 15%, compared to patients who remained abstinent.”

4. Baseline data: Again it is not very clear but important for this study how patients were enrolled: It reads that “Patients with harmful alcohol consumption and evidence of alcohol-dependence, followed in an Alcohol Rehabilitation Clinic, were prospectively and consecutively enrolled in a clinical protocol performed in 2002 ref 13 and retrospectively re-evaluated in 2018. What means retrospectively in this context? How was liver disease excluded?

We added the following statement, to the methods, regarding baseline data: “At enrollment all patients attending outpatient visits at Centro Regional de Alcoologia do Sul (CRAS) were asked to participate in the study and were evaluated at an Hepatology ambulatory clinic with laboratorial evaluation.”

The retrospective evaluation is explained in the methods: “All data were obtained from the computerized national health database (including hospital admissions or consultations, primary care appointments, blood tests, and imaging results), from consultation of the records in the Alcohol Rehabilitation Clinic at 2011, and through a survey performed by telephone interview with the patient or patient´s relatives.” At the end of follow up, liver disease was excluded as explained: “Patients were considered to have developed CLD when they presented abnormal liver tests for more than 6 months, imaging findings compatible with CLD, or presented with hepatic decompensation (jaundice, ascites, hepatic encephalopathy, and/or variceal bleeding).”

5. I am also confused when looking at Ref 13 (Martins A et al Eur J Gastroenterol Hepatol 2005; 17 (10):1099-1104.). In this study, completely different patient numbers are provided.

This cohort of patients was enrolled at 2002, as part of a study regarding genetic susceptibility for alcohol-associated liver disease. For the manuscript Martins A et al Eur J Gastroenterol Hepatol 2005; 17 (10):1099-1104, only 76 patients were included in the group of heavy drinkers without chronic liver disease, because for inclusion we needed two laboratorial evaluations with normal liver disease and many patients did not have available other laboratorial evaluation apart from the one performed at recruitment. In the following year, many patients evaluated did perform a second laboratorial evaluation 6 months apart with normal liver tests. As such, the pool of heavy drinkers without evidence of chronic liver disease increased. We referenced that manuscript because it is the same cohort, with the same methodology. However, for clarity, we have changed to “with data partially reported in 2005 [13]”

6. It is also interesting that obviously an isolated rise in GGT was not considered an exclusion criterium. Why? How many patients were enrolled based on these criterium.

We did not exclude patients with isolated rise in GGT, because we consider that GGT in this setting may be only a marker of alcohol consumption, rather than alcohol-associated liver disease. Striking elevations of serum GGT activities can be observed in patients with a high alcohol intake over a prolonged period, in patients without liver disease other than fatty liver. The increase in GGT seems to be primarily due to hepatic enzyme induction in the endoplasmic reticulum, rather than to liver cell injury. Isolated increase in GGT was present in 39 patients (31.7%). Interestingly, higher GGT levels did not associate with an increased risk of progression to chronic liver disease.

To clarify this, we added the following sentence to the methods: “Isolated increase of GGT was not an exclusion criteria, since it is a marker of alcohol consumption and not of liver injury, translating hepatic enzyme induction rather than liver cell injury {Nishimura, 1983 #55}. Isolated increase of GGT was found in 39 patients (31.7%).”

7. It is further pointed out that follow up data were retrospectively obtained form a national database. IT would be interesting to learn how data on alcohol consumption are regularly entered in this database. Who is doing this? How often did patients undergo alcohol detoxification?

8. Since alcohol drinking history is essential for this study, more details should be provided, given the fact that it is generally challenging for patients to provide a detailed alcohol history with stops and goes. How was duration of alcohol consumption assessed? How do patients remember an alcohol drinking history of the last 10 years or so? In line with this, a 53% of relapse appears to be rather low for heavy drinkers.

The national database consists of the outpatients visits in primary care, as well as outpatient visits and hospitalizations in public hospitals. Working in a public hospital, a physician has access to the medical records from other public institutions, regarding the patient they are following or did follow. Because all patients were evaluated in a Hepatology visit in our institution, we did have access to their medical records. We do realize that information regarding alcohol consumption in medical records most of the times is not thorough. We also, did consult the medical records of the alcohol rehabilitation clinic at 2011. As such, the history of alcohol intake during follow up was collected by a thorough questionnaire to the patient and their relatives, confronted with the information collected from medical records. We do recognize that there may be recall issues in the assessment of alcohol consumption, and that was now emphasized in the discussion. Accordingly, we added to the discussion: “Still, the current study also has some limitations. First, the risk of bias due to the nature of alcohol history questionnaires with recall issues. Indeed, we tried to be thorough in the evaluation of alcohol consumption, applying a structured questionnaire to patients and their relatives, while confronting them with the information collected from medical records from the national database, as well as information collected from the medical records from the alcohol rehabilitation clinical at the middle of the follow up.”

We did not register how often patients did undergo alcohol detoxification.

All patients were followed in an alcohol rehabilitation clinic, which may explain the rather low percentage of relapse in those heavy drinkers.

9. On page 12 it is stated that 9% had previous AH while in the inclusion criteria, previous decompensation was ruled out?

We did not account for asymptomatic ambulatory alcohol hepatitis pattern as hepatic decompensation. We amended the phrase to: “9% had previous biochemical changes compatible with asymptomatic ambulatory alcoholic hepatitis”

10. What about the numbers of diabetes and obesity?

We added the following to the results: “Eighty-one (66%) were smokers, 48 (46%) overweight/obese (28 obese, 23%) and 6 patients had type 2 diabetes mellitus (5%)”.

11. Smoking is shown as risk factor for DM II with an OR 0.181 Does smoking protect from diabetes? There are studies continuously showing a tight association between smoking and alcohol consumption? What are the numbers for this cohort?

We did found an inverse association between baseline smoking and the development of type 2 diabetes mellitus, which is not in agreement with literature. Of note, we do not have information regarding smoking status on follow up, nor the number of cigarettes smoked per day, and that may explain these results. This is now stressed in the results: “Of note, we did not have information regarding the smoking status on follow up.” Baseline smoking did not associate with alcohol consumption relapse.

REVIEWER 2

This is an interesting prospective study that studies the natural history of heavy drinkers without evidence of liver disease and evaluates their clinical outcomes, risk of death and of developing liver disease during 15 years of follow up. Risk factors were also identified.

From this standpoint, the study is important because little is know about the risk of developing CLD in heavy drinkers. However, there are some issues that need further clarification.

We thank the reviewer for these encouraging words.

1. The data in table 1 are presented as mean±SD, which - based on the figures, seem not quite appropriate (at least for some of the variables. Could you please look back to the distribution of the variables and use median±IQR/range when needed. Also, in some cases, non parametric tests should have been used.

We thank the reviewer for this pertinent comment. We changed it accordingly.

2. Regarding the data presented in figure 1, I think that the AUROC analysis is not the right choice for what was intended by the authors. AUROC is usually used with continuous variables, not with dichotomial ones. I believe a regression analysis (or a basic chi-square test) would be better choices.

We agree with the reviewer, and we changed it accordingly.

3. The manuscript also need a further English polish it increase its readability.

We carefully edited the text to improve readability.

REVIEWER 3

Mariana Machado and colleagues present a single-center study from Lisbon on the natural history of 130 patients with a history of excessive drinking who present at an alcohol rehabilitation unit, but without evidence of chronic liver disease or cancer. The authors are commended for managing to recall almost all (7 losses to follow up) participants after 14±3 years for repeated investigations and a detailed description of alcohol use during follow up.

While there is a need for longitudinal studies on the natural history of patients with harmful drinking, taking fluctuations in alcohol use over the years into account, the study in its current form has substantial methodological and reporting issues which limits interpretation of results. Please find below my questions, comments and suggestions for improvement.

We thank the reviewer for these encouraging words.

1. Please use the STROBE guidelines for reporting: https://www.equator-network.org/reporting-guidelines/strobe/

We thank the reviewer for this comment. We made the necessary changes to follow the STROBE guidelines for reporting:

 Item No Recommendation Page No

Title and abstract 1 (a) Indicate the study’s design with a commonly used term in the title or the abstract 3

 (b) Provide in the abstract an informative and balanced summary of what was done and what was found 

Introduction

Background/rationale 2 Explain the scientific background and rationale for the investigation being reported 6

Objectives 3 State specific objectives, including any prespecified hypotheses 7

Methods

Study design 4 Present key elements of study design early in the paper 8

Setting 5 Describe the setting, locations, and relevant dates, including periods of recruitment, exposure, follow-up, and data collection 8

Participants 6 (a) Give the eligibility criteria, and the sources and methods of selection of participants. Describe methods of follow-up 8

 (b) For matched studies, give matching criteria and number of exposed and unexposed n.a.

Variables 7 Clearly define all outcomes, exposures, predictors, potential confounders, and effect modifiers. Give diagnostic criteria, if applicable 9,10

Data sources/ measurement 8* For each variable of interest, give sources of data and details of methods of assessment (measurement). Describe comparability of assessment methods if there is more than one group 9,10

Bias 9 Describe any efforts to address potential sources of bias 10

Study size 10 Explain how the study size was arrived at n.a.

Quantitative variables 11 Explain how quantitative variables were handled in the analyses. If applicable, describe which groupings were chosen and why 11

Statistical methods 12 (a) Describe all statistical methods, including those used to control for confounding 11

 (b) Describe any methods used to examine subgroups and interactions n.a.

 (c) Explain how missing data were addressed n.a.

 (d) If applicable, explain how loss to follow-up was addressed 8

 (e) Describe any sensitivity analyses 12

Results 

Participants 13* (a) Report numbers of individuals at each stage of study—eg numbers potentially eligible, examined for eligibility, confirmed eligible, included in the study, completing follow-up, and analysed 12

 (b) Give reasons for non-participation at each stage 12

 (c) Consider use of a flow diagram n.a.

Descriptive data 14* (a) Give characteristics of study participants (eg demographic, clinical, social) and information on exposures and potential confounders Table 1

 (b) Indicate number of participants with missing data for each variable of interest n.a.

 (c) Summarise follow-up time (eg, average and total amount) 12

Outcome data 15* Report numbers of outcome events or summary measures over time 13

Main results 16 (a) Give unadjusted estimates and, if applicable, confounder-adjusted estimates and their precision (eg, 95% confidence interval). Make clear which confounders were adjusted for and why they were included Table 1

 (b) Report category boundaries when continuous variables were categorized n.a.

 (c) If relevant, consider translating estimates of relative risk into absolute risk for a meaningful time period n.a.

Other analyses 17 Report other analyses done—eg analyses of subgroups and interactions, and sensitivity analyses n.a.

Discussion

Key results 18 Summarise key results with reference to study objectives 17

Limitations 19 Discuss limitations of the study, taking into account sources of potential bias or imprecision. Discuss both direction and magnitude of any potential bias 20

Interpretation 20 Give a cautious overall interpretation of results considering objectives, limitations, multiplicity of analyses, results from similar studies, and other relevant evidence 20

Generalisability 21 Discuss the generalisability (external validity) of the study results 20,21

Other information

Funding 22 Give the source of funding and the role of the funders for the present study and, if applicable, for the original study on which the present article is based 2

2. Title: the short titles accurately states that development of CLD during follow up is the outcome of interest, while the long title does not.

We changed the title to: “Outcomes of excessive alcohol drinkers without baseline evidence of chronic liver disease after 15 years follow-up: heavy burden of cancer and liver disease mortality”

3. Lay summary, first point. It is not surprising that ongoing rather than past alcohol intake dictates the risk of liver cirrhosis development when you specifically exclude anyone with evidence of liver disease at baseline. Therefore I suggest to delete this sentence.

Thank you for the suggestion. Accordingly, we changed the sentence to: ongoing alcohol, and deleted: “rather than past”

4. The authors stress the point that alcohol consumers on average has a 10 yr reduced life expectancy. Their way of comparing cohort data with a country average is not really methodologically correct – the correct comparator would be 5:1 or 10:1 gender, age, time and geography matched random sample from the population. Therefore, I would propose not to emphasize the 10 yr reduction in life expectancy as much as the authors do – e.g. it appears in both abstract and bullet points.

We agree with the reviewer, as such we removed that sentence from the lay summary. In the abstract we reduced the emphasis of the statement, from “showed an impressive decrease” to “seemed to have a decreased”.

5. Introduction: The statement “Liver disease is the main alcohol-related chronic illness” gives the wrong impression, since liver disease – as the authors show – does not account for the majority of alcohol-related harm. I would propose instead to stress that the liver is the single organ most commonly affected by excess drinking.

We thank the reviewer for this comment, and we changed it accordingly.

6. Methods: Reference 13 is referred to as the original study. However, it describes a cohort of only 76 patients without evidence of liver disease at baseline, but this manuscript includes n=130?

This cohort of patients was enrolled at 2002, as part of a study regarding genetic susceptibility for alcohol-associated liver disease. For the manuscript Martins A et al Eur J Gastroenterol Hepatol 2005; 17 (10):1099-1104, only 76 patients were included in the group of heavy drinkers without chronic liver disease, because for inclusion we needed two laboratorial evaluations with normal liver disease and many patients did not have available other laboratorial evaluation apart from the one performed at recruitment. In the following year, many patients evaluated did perform a second laboratorial evaluation 6 months apart with normal liver tests. As such, the pool of heavy drinkers without evidence of chronic liver disease increased. We referenced that manuscript because it is the same cohort, with the same methodology. For clarity, we have changed to “with data partially reported in 2005 [13]”.

7. Methods: Please describe in more detail the in- and exclusion criteria. Which liver blood tests were considered? If fibrosis stage F1 on liver biopsy, the patient were considered to have ALD and excluded? From the text is seem like a very heterogeneous cohort, since many with compensated cirrhosis at baseline may have normal liver blood tests (see e.g. Table 1 in Mueller, World J Gastroenterol 2014 October 28; 20(40): 14626-14641). Those patients would be included unless they had a liver biopsy? In contrast, many with no or minimal fibrosis may have elevated transaminases and alkaline phosphatase – they would be excluded?

We added the following to the methods, which we hope clarifies the inclusion and exclusion criteria: “Requisites for the absence of CLD included: the absence of a history of hepatic decompensation, absence of physical stigmata of chronic liver disease (i.e. cutaneous signs, hepatosplenomegaly, gynecomastia, and testicular atrophy), and at least two normal determinations of blood liver tests (aminotransferases, alkaline phosphatase, total bilirubin, protrhombin time and albumin) with an exception of an isolated rise in γ-glutamyl transpeptidase (GGT). Of the 262 screened patients, 130 did not have evidence of CLD, 103 had alcohol-associated liver disease, and the remaining 29 had either other liver diseases or mild increase in liver enzymes (that is, an increase lower than 3 times the reference value). Isolated increase of GGT was not an exclusion criteria, since it is a marker of alcohol consumption and not of liver injury, translating hepatic enzyme induction rather than liver cell injury{Nishimura, 1983 #55}. Isolated increase of GGT was found in 39 patients (31.7%) without evidence of liver disease. Patients who underwent liver biopsy raised clinical doubts regarding the presence of liver disease, and hence were not classified as not having liver disease even if the biopsy was normal, because of concerns of sampling error.

8. Methods: Similarly, the definition of the primary outcome, development of CLD, is poorly described. The current description can encompass patients with anything from steatohepatitis without fibrosis, all the way to decompensated cirrhosis.

We agree with the reviewer, one limitation of the study is the retrospective design of the follow up. As such, we considered clinically evident chronic liver disease, either by persistent abnormal liver tests, imaging evidence of chronic liver disease or clinical decompensation of liver disease, as stated in the methods.

9. Methods: The authors state that inclusions were in 2002 and reassessment in 2018, which would result in 16 years between baseline and reevaluation. Yet, in methods they state a follow up of 14 ±3 yrs?

Follow up was smaller than 16 years because many patients died, thus reducing the mean follow up.

10. Clearly, the alcohol habits of those who die in between baseline and follow up can not be evaluated, or did the authors have information from other sources on alcohol behaviour during follow up?

Thank you for the observation. We agree, and added the following sentence: “Indeed, we tried to be thorough in the evaluation of alcohol consumption, applying a structured questionnaire to patients and their relatives, while confronting them with the information collected from medical records from the national database, as well as information collected from the medical records from the alcohol rehabilitation clinical at the middle of the follow up.”

11. Methods: When the primary endpoint is risk of CLD, the authors should account for how they handle non-CLD death as a competing risk?

We did that sensitive analysis by performing Kaplan-Meier curves of CLD-free survival, with Cox-regression for confounders.

12. Methods: Baseline CLD and cancer are excluded, but cardiovascular or metabolic disease are not, so the comparison between deaths from CLD, cancer and CVD is unbalanced.

We agree with the reviewer, the comparison may be unbalanced, but that would favor CVD. However, we did find unexpectedly high CLD and cancer mortality, and not CVD, which further strengthen the burden of CLD and cancer mortality.

13. Results: seven patients (6%) progressed to CLD, which is lower than in studies of similar populations (for example Holst, Addiction 2017, vol 112, where approximately 25% die from liver related death). Could the authors discuss this? Would more accurate liver tests – like FibroScan, which was widely available in 2018 – have resulted in higher proportion being detected with CLD?

Thank you for your observation. We believe the low risk of progression to CLD can be explained by the fact that this is a highly selected population, of patients without evidence of CLD baseline, even though with a median alcohol consumption of 271±203g/day during 21.5±20 years. This cohort represents a population particularly resistant to CLD, and even so, 6% still progress to CLD. The study mentioned by the reviewer does not exclude patients already with CLD when referred to an alcohol rehabilitation clinic, which might explain the higher liver-related death rate. In accordance we have now added the phrase: Interestingly, our cohort had a lower rate of progression (6% vs 25%) than the Danish cohort, probably related to the fact that these are patients without evidence of CLD baseline, suggesting a significant degree of resistance to liver disease development.

14. Results: It makes less sense to discuss biomarkers of liver disease, when abnormal liver blood tests were an exclusion criterion.

We understand the reviewer’s point, however we believe this analysis is still relevant. First of all, we did accept isolated increase in GGT, and the fact that it did not associate with an increased risk for progression to CLD validates our choice. Second, even though a parameter is in the normal range, differences in the normal range may still have subtle impact in the risk of disease. One such example is the impact of body mass index in the development of hepatic steatosis. We know that even in the normal range, the higher the BMI, the higher the risk of developing hepatic steatosis. Indeed, we did find that the higher the alkaline phosphatase, the higher the risk of developing chronic liver disease, which may be explained by subclinical cholestasis.

15. Discussion: It is problematic that there is such a long time period between baseline and follow up with regards to obtaining alcohol history. I fear that recall bias may play a strong role, since those who have developed CLD will be more prone to remember alcohol drinking periods than those who have not. Can the authors discuss this limitation?

We agree, we discussed it: First, the risk of bias due to the nature of alcohol history questionnaires with recall issues. Indeed, we tried to be thorough in the evaluation of alcohol consumption, applying a structured questionnaire to patients and their relatives, while confronting them with the information collected from medical records from the national database, as well as information collected from the medical records from the alcohol rehabilitation clinical at the middle of the follow up.”

16. I recommend a language revision by a native English speaker. Examples from the abstract: “Most long-term heavy drinkers do not present chronic liver disease” (present with CLD?); “being abstinent for at least 1 year positively modeled CLD-free survival” (correlated or associated with CLD-free survival?); “showed an impressive decreased life expectancy” (impressive has positive connotations, which is in stark contrast to decreased life expectancy).

We carefully edited the text to improve readab

---

## [Decision Letter · Decision Letter 1]

30 Apr 2021

PONE-D-21-05110R1

Outcomes of excessive alcohol drinkers without baseline evidence of chronic liver disease after 15 years follow-up: heavy burden of cancer and liver disease mortality

PLOS ONE

Dear Dr. Machado,

Thank you for submitting your manuscript to PLOS ONE. After careful consideration, we feel that it has merit but does not fully meet PLOS ONE’s publication criteria as it currently stands. Therefore, we invite you to submit a revised version of the manuscript that addresses the points raised during the review process.

Please follow the recommendations of Reviewer #3 in order to further improve the presentation of your interesting findings.

The abbreviation 'LC' should be replaced by 'cirrhosis' (not 'liver cirrhosis' as there is no other cirrhosis). 

We look forward to receiving your revised manuscript.

Kind regards,

Rudolf E. Stauber, MD

Academic Editor

PLOS ONE

Journal Requirements:

Reviewers' comments:

Reviewer's Responses to Questions

**Comments to the Author**

1. If the authors have adequately addressed your comments raised in a previous round of review and you feel that this manuscript is now acceptable for publication, you may indicate that here to bypass the “Comments to the Author” section, enter your conflict of interest statement in the “Confidential to Editor” section, and submit your "Accept" recommendation.

Reviewer #1: All comments have been addressed

Reviewer #2: All comments have been addressed

Reviewer #3: All comments have been addressed

2. Is the manuscript technically sound, and do the data support the conclusions?

Reviewer #1: Yes

Reviewer #2: Yes

Reviewer #3: Yes

3. Has the statistical analysis been performed appropriately and rigorously? 

Reviewer #1: Yes

Reviewer #2: Yes

Reviewer #3: Yes

4. Have the authors made all data underlying the findings in their manuscript fully available?

Reviewer #1: Yes

Reviewer #2: Yes

Reviewer #3: No

5. Is the manuscript presented in an intelligible fashion and written in standard English?

Reviewer #1: Yes

Reviewer #2: Yes

Reviewer #3: No

6. Review Comments to the Author

Reviewer #1: I am satisfied with the author's response and modifications of the manuscript.

Acceptance recommended.

Reviewer #2: (No Response)

Reviewer #3: Thank you for the revised version of this manuscript. I have just a few additional suggestions and comments:

General:

• Please throughout the manuscript state both absolute numbers and proportions. The cohort is relatively small, so the sole reporting of percentages gives a wrong impression.

• Language: English editing is still needed. There remain grammar errors and sentence structures which gives a poor impression, and at certain places makes the text difficult to even understand.

• LC is not a common abbreviation; I suggest to write liver cirrhosis in full throughout.

Results:

- There is a lack of time to progression for all outcomes, except for the Kaplan Meier plots. This hinders accurate interpretation, and should be stated in the text.

- Figure 1: The ROC analysis is not the correct method to assess prognostic accuracy as it is not time-dependent. Should be replaced with Harrell’s C or time-dependent ROC curves.

Discussion:

- Based on the low event rate and relatively high survival for this type of patient I still suspect a high degree of survivors bias. The authors correctly state: “The first striking finding, from this cohort, is that patients who have no evidence of CLD despite being heavy-drinkers for 25 years are still at increased risk for CLD”. I suggest to more clearly acknowledge and discuss the limitation of survivors bias and that this is not an inception cohort. Rather, included participants have already survived >20 years of heavy drinking without developing evidence of chronic liver disease and this population is therefore selected to be less likely to develop liver disease during the next 14 years, than the average patients with heavy drinking.

- Regarding excess cancer risk due to combined alcohol and smoking. The sentence “This population should probably be submitted to a personalized screening program.” is unsubstantiated as this cohort is not suited to investigate such topic.

- Similarly the discussion about beer drinking having a possible protective effect is purely speculative and this study is not powered or designed to make such assumptions. The effect of beer may also be that it is the type of beverage containing the lowest percentage of alcohol; or just a common type I error.

- The discussion of lack of liver biopsy may be replaced with lack of accurate confirmatory tests – LB or LSM – to ensure the absence of liver disease.

- I am not sure I understand the sentence “These patients represent the largest subgroup of heavy-drinkers on clinical practice”. I guess it depends on which healthcare setting one operates in?

7. PLOS authors have the option to publish the peer review history of their article (what does this mean?). If published, this will include your full peer review and any attached files.

Reviewer #1: No

Reviewer #2: No

Reviewer #3: No

---

## [Author Response · Author response to Decision Letter 1]

9 May 2021

RESPONSE TO THE REVIEWERS

We thank the editor and the reviewers for the constructive critiques and for the opportunity to improve our manuscript. We hope the reviewers now find it worthy for publication. It follows the point-by-point response to the editors and reviewers comments.

RESPONSE TO REVIEWER 3

Thank you for the revised version of this manuscript.

I have just a few additional suggestions and comments:

General:

• Please throughout the manuscript state both absolute numbers and proportions. The cohort is relatively small, so the sole reporting of percentages gives a wrong impression.

R: It was changed accordingly.

• Language: English editing is still needed. There remain grammar errors and sentence structures, which gives a poor impression, and at certain places makes the text difficult to even understand.

R: The English was carefully edited.

• LC is not a common abbreviation; I suggest to write liver cirrhosis in full throughout.

R: Thank you for the suggestion. It was changed accordingly.

Results:

• There is a lack of time to progression for all outcomes, except for the Kaplan Meier plots. This hinders accurate interpretation, and should be stated in the text.

R: It was stated in the text: “In order to allow for an analysis of time to progression, Kaplan Meyer curves were computed.” We also added in the discussion: “Lastly, the analysis of the effect of variables and confounders on outcomes has the risk of bias by the lack of control to time to progression, which was taken into account in the evaluation of the survival curves.”

• Figure 1: The ROC analysis is not the correct method to assess prognostic accuracy as it is not time-dependent. Should be replaced with Harrell’s C or time-dependent ROC curves.

R: We agree that ROC analysis was not adequate to evaluate the abstinence time, as such we removed it in the previous revision of the manuscript.

Discussion:

• Based on the low event rate and relatively high survival for this type of patient I still suspect a high degree of survivors bias. The authors correctly state: “The first striking finding, from this cohort, is that patients who have no evidence of CLD despite being heavy-drinkers for 25 years are still at increased risk for CLD”. I suggest to more clearly acknowledge and discuss the limitation of survivors bias and that this is not an inception cohort. Rather, included participants have already survived >20 years of heavy drinking without developing evidence of chronic liver disease and this population is therefore selected to be less likely to develop liver disease during the next 14 years, than the average patients with heavy drinking.

This is the main interest of this cohort. Albeit it is a highly selected cohort of patients less likely to develop CLD, since they already survived more than 20 years of heavy drinking without developing evidence of CLD, 6% still progresses. The risk of progression is not related with past alcohol consumption, rather than on ongoing alcohol intake.

R: We changed the sentence to: “The first striking finding, from this cohort, is that patients who survived up to 25 years of heavy-drinking without developing evidence of CLD, and hence who were more likely resistant to the dismal effects of alcohol on the liver, are still at increased risk for CLD.”

• Regarding excess cancer risk due to combined alcohol and smoking. The sentence “This population should probably be submitted to a personalized screening program.” is unsubstantiated as this cohort is not suited to investigate such topic.

R: We agree with the reviewer, and as such, we changed the sentence to: “Further studies should address if this population should be submitted to a personalized screening program.”

• Similarly the discussion about beer drinking having a possible protective effect is purely speculative and this study is not powered or designed to make such assumptions. The effect of beer may also be that it is the type of beverage containing the lowest percentage of alcohol; or just a common type I error.

R: We removed the paragraph regarding the discussion about the possible protective effect of beer drinking.

• The discussion of lack of liver biopsy may be replaced with lack of accurate confirmatory tests – LB or LSM – to ensure the absence of liver disease.

R: Accordingly, we added the following sentences: “Furthermore, transient hepatic elastography was also not performed because it was not accessible at the time of inclusion of the patients. However, liver biopsy is not routinely performed in these patients in the real-life clinical practice, and many centers (mainly in the primary care set) still do not have easy access to transient hepatic elastography.”

• I am not sure I understand the sentence “These patients represent the largest subgroup of heavy-drinkers on clinical practice”. I guess it depends on which healthcare setting one operates in?

R: We clarified the sentence as following: “These patients represent the largest subgroup of heavy-drinkers on clinical practice (particularly in Primary Care Clinics)”

---

## [Editor Report · Decision Letter 2]

12 May 2021

Outcomes of excessive alcohol drinkers without baseline evidence of chronic liver disease after 15 years follow-up: heavy burden of cancer and liver disease mortality

PONE-D-21-05110R2

Dear Dr. Machado,

We’re pleased to inform you that your manuscript has been judged scientifically suitable for publication and will be formally accepted for publication once it meets all outstanding technical requirements.

Kind regards,

Rudolf E. Stauber, MD

Academic Editor

PLOS ONE

Additional Editor Comments (optional):

The issues raised by Reviewer 3 have been answered satisfactorily.

Page 18: please correct Kaplan Meyer  Kaplan-Meier. 
---

## [Editor Report · Acceptance letter]

17 May 2021

PONE-D-21-05110R2 

Outcomes of excessive alcohol drinkers without baseline evidence of chronic liver disease after 15 years follow-up: heavy burden of cancer and liver disease mortality 

Dear Dr. Machado:

I'm pleased to inform you that your manuscript has been deemed suitable for publication in PLOS ONE. Congratulations! Your manuscript is now with our production department. 

Kind regards, 

on behalf of

Professor Rudolf E. Stauber 

Academic Editor

PLOS ONE